# SIMPLEKT: A SIMPLE BUT TOUGH-TO-BEAT BASELINE FOR KNOWLEDGE TRACING

**Zitao Liu**
Guangdong Institute of Smart Education, Jinan University, Guangzhou, China
`liuzitao@jnu.edu.cn`

**Qiongqiong Liu, Jiahao Chen, Shuyan Huang**[*]
TAL Education Group, Beijing, China
`{liuqiongqiong1, chenjiahao, huangshuyan}@tal.com`

**Weiqi Luo**
Guangdong Institute of Smart Education, Jinan University, Guangzhou, China
`lwq@jnu.edu.cn`

## ABSTRACT

Knowledge tracing (KT) is the problem of predicting students' future performance based on their historical interactions with intelligent tutoring systems. Recently, many works present lots of special methods for applying deep neural networks to KT from different perspectives like model architecture, adversarial augmentation and etc., which make the overall algorithm and system become more and more complex. Furthermore, due to the lack of standardized evaluation protocol (Liu et al., 2022), there is no widely agreed KT baselines and published experimental comparisons become inconsistent and self-contradictory, i.e., the reported AUC scores of DKT on ASSISTments2009 range from 0.721 to 0.821 (Minn et al., 2018; Yeung & Yeung, 2018). Therefore, in this paper, we provide a strong but simple baseline method to deal with the KT task named SIMPLEKT. Inspired by the Rasch model in psychometrics, we explicitly model question-specific variations to capture the individual differences among questions covering the same set of knowledge components that are a generalization of terms of concepts or skills needed for learners to accomplish steps in a task or a problem. Furthermore, instead of using sophisticated representations to capture student forgetting behaviors, we use the ordinary dot-product attention function to extract the time-aware information embedded in the student learning interactions. Extensive experiments show that such a simple baseline is able to always rank top 3 in terms of AUC scores and achieve 57 wins, 3 ties and 16 loss against 12 DLKT baseline methods on 7 public datasets of different domains. We believe this work serves as a strong baseline for future KT research. Code is available at `https://github.com/pykt-team/pykt-toolkit`[1].

## 1 INTRODUCTION

Knowledge tracing (KT) is a sequential prediction task that aims to predict the outcomes of students over questions by modeling their mastery of knowledge, i.e., knowledge states, as they interact with learning platforms such as massive open online courses and intelligent tutoring systems, as shown in Figure 1. Solving the KT problems may help teachers better detect students that need further attention, or recommend personalized learning materials to students.

The KT related research has been studied since 1990s where Corbett and Anderson, to the best of our knowledge, were the first to estimate students' current knowledge with regard to each individ-

---

[*]The corresponding author: Shuyan Huang.
[1]We merged our model to the PYKT benchmark at `https://pykt.org/`

ual knowledge component (KC) (Corbett & Anderson, 1994). A KC is a description of a mental structure or process that a learner uses, alone or in combination with other KCs, to accomplish steps in a task or a problem[2]. Since then, many attempts have been made to solve the KT problem, such as probabilistic graphical models (Käser et al., 2017) and factor analysis based models (Cen et al., 2006; Lavoué et al., 2018; Thai-Nghe et al., 2012). Recently, with the rapid development of deep neural networks, many deep learning based knowledge tracing (DLKT) models are developed, such as auto-regressive based deep sequential KT models (Piech et al., 2015; Yeung & Yeung, 2018; Chen et al., 2018; Wang et al., 2019; Guo et al., 2021; Long et al., 2021; Chen et al., 2023), memory-augmented KT models (Zhang et al., 2017; Abdelrahman & Wang, 2019; Yeung, 2019), attention based KT models (Pandey & Karypis, 2019; Pandey & Srivastava, 2020; Choi et al., 2020; Ghosh et al., 2020; Pu et al., 2020), and graph based KT models (Nakagawa et al., 2019; Yang et al., 2020; Tong et al., 2020).

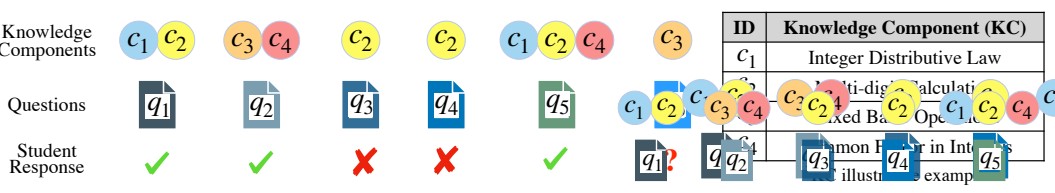

Figure 1: Graphical illustration of the task of Knowledge Tracing. "✓" and "✗" denote the question is answered correctly and incorrectly.

Although DLKT approaches have constituted new paradigms of the KT problem and achieved promising results, recently developed DLKT models seem to be more and more complex and resemble each other with very limited nuances from the methodological perspective: applying different neural components to capture student forgetting behaviors (Ghosh et al., 2020; Nagatani et al., 2019), recency effects (Zhang et al., 2021), and various auxiliary information including relations between questions and KCs (Tong et al., 2020; Pandey & Srivastava, 2020; Liu et al., 2021; Yang et al., 2020), question text content (Liu et al., 2019; Wang et al., 2020a), question difficulty level (Liu et al., 2021; Shen et al., 2022), and students' learning ability (Shen et al., 2020). Furthermore, published DLKT baseline results surprisingly diverge. For example, the reported AUC scores of DKT and AKT on ASSISTments2009 range from 0.721 to 0.821 (Minn et al., 2018; Yeung & Yeung, 2018) and from 0.747 to 0.835 in (Ghosh et al., 2020; Wang et al., 2021) respectively. Another example is that for the performance of DKT on the ASSISTments2009 dataset, it is recognized as one of the best baselines by Ghosh et al. (2020) while Long et al. (2022) and Zhang et al. (2021) showed that its performance is below the average. Recent survey studies by Sarsa et al. (2022) and Liu et al. (2022) summarized aforementioned inconsistencies of baseline results and showed evidence that variations in hyper-parameters and data pre-processing procedures contribute significantly to prediction performance of DLKT models. Specifically, Sarsa et al. (2022) empirically found that even simple baselines with little predictive value may outperform DLKT models with sophisticated neural components. Liu et al. (2022) built a standardized DLKT benchmark platform and showed that the improvement of many DLKT approaches is minimal compared to the very first DLKT model proposed by Piech et al. (2015).

Therefore, in this paper, we propose SIMPLEKT, a simple but tough-to-beat KT baseline that is simple to implement, computationally friendly and robust to a wide range of KT datasets across different domains. Motivated by the Rasch model[3] that is a classic yet powerful model in psychometrics, the proposed SIMPLEKT approach captures the individual differences among questions covering the same set of KCs by representing each question's embedding as an additive combination of the average of its corresponding KCs' embeddings and a question-specific variation. Furthermore, different from many existing models that try to capture various aforementioned relations and information, the SIMPLEKT is purely based on the attention mechanism and uses the ordinary dot-product attention function to capture the contextual information embedded in the student learning interactions. To comprehensively and systematically evaluate the performance of SIMPLEKT, we choose to use the publicly available PYKT benchmark[4] implementation to guarantee valid and reproducible comparisons against 12 DLKT methods on 7 popular datasets across differ-

---

[2]A KC is a generalization of everyday terms like concept, principle, fact, or skill.

[3]The Rasch model is also known as the 1PL item response theory model.

[4]https://www.pykt.org/

ent domains. Results shown that the SIMPLEKT beats a wide range of modern neural KT models that based on graph neural networks, memory augmented neural networks, and adversarial neural networks. This suggests that this simple method should be used as the baseline to beat future KT research, especially when designing sophisticated neural KT architectures. To encourage reproducible research, all the related codes, data and the learned SIMPLEKT models are publicly available at `https://github.com/pykt-team/pykt-toolkit`.

## 2 RELATED WORK

Recently, deep learning technique have been widely applied into KT task for student's historical learning modeling and the future performance prediction. Existing DLKT approaches can be categorized into the following 5 categories:

**C1: Deep sequential models**. DLKT models that use auto-regressive architectures to capture students' chronologically ordered interactions. For example, (Piech et al., 2015) proposed the very first DKT model that utilizes an LSTM layer to estimate the knowledge mastery. (Lee & Yeung, 2019) proposed to enhance DKT with a skill encoder that combines student learning activities and KC representations.

**C2: Memory augmented models**. DLKT models that capture latent relations between KCs and student knowledge states via memory networks. For instance, (Zhang et al., 2017) exploited and stored the KC relationships via a static key memory matrix and predict students' knowledge mastery levels with a dynamic value memory matrix.

**C3: Adversarial based models**. DLKT models that utilize the adversarial techniques to generate perturbations to improve model generalization capability. (Guo et al., 2021) jointly train an attentive-LSTM KT model with both original and adversarial examples.

**C4: Graph based models**. DLKT models that use the graph neural networks to model intrinsic relations among questions, KCs and interactions. (Liu et al., 2021) presented a question-KC bipartite graph to explicitly capture question-level and KC-level inner-relations and question difficulties. (Yang et al., 2020) introduced a graph convolutional network to obtain the representation of the question-KC correlations.

**C5: Attention based models**. DLKT models that capture dependencies between interactions via the attention mechanism. For example, (Pandey & Karypis, 2019) used self-attention network to capture the relevance between KCs and students' historical interactions. Choi et al. (2020) designed an encoder-decoder structure to represent the exercise and response embedding sequences. (Ghosh et al., 2020) performed three self-attention modules and explicitly model students' forgetting behaviors via a monotonic attention mechanism.

Please note that the above categorizations are not exclusive and related techniques can be combined. For example, (Abdelrahman & Wang, 2019) proposed a sequential key-value memory network to unify the strengths of recurrent modeling capacity and memory capacity. The proposed SIMPLEKT approach belongs to C5 and it purely models student interactions by using the very ordinary dot-product attention function.

## 3 A SIMPLE METHOD FOR KNOWLEDGE TRACING

### 3.1 PROBLEM STATEMENT

In this work, our objective is given an arbitrary question $q_*$ to predict the probability of whether a student will answer $q_*$ correctly or not based on the student's historical interaction data. More specifically, for each student $\mathbf{S}$, we assume that we have observed a chronologically ordered collection of $T$ past interactions i.e., $\mathbf{S} = \{\mathbf{s}_j\}_{j=1}^{T}$. Each interaction is represented as a 4-tuple $\mathbf{s}$, i.e., $\mathbf{s} =< q, \{c\}, r, s >$, where $q, \{c\}, r, s$ represent the specific question, the associated KC set, the binary valued student response[5], and student's response time step respectively. We would like to estimate the probability $\hat{r}_*$ of the student's performance on arbitrary question $q_*$.

---

[5]Response is a binary valued indicator variable where 1 represents the student correctly answered the question, and 0 otherwise.

## 3.2 THE SIMPLEKT APPROACH

### 3.2.1 REPRESENTATIONS OF QUESTIONS, KCS AND RESPONSES.

Effectively representing student interactions is crucial to the success of the DLKT models. In real-world educational scenarios, the question bank is usually much bigger than the set of KCs. For example, the number of questions is more than 1500 times larger than the number of KCs in the Algebra2005 dataset (described in Section 4.1). Therefore, to effectively learn and fairly evaluate the DLKT models from such highly sparse question-response data, following the previous work of (Ghosh et al., 2020) and (Liu et al., 2022), we artificially transform the original question-response data into KC-response data by expanding each question-level interaction into multiple KC-level interactions when the question is associated with a set of KCs (illustrated in Figure 2).

Furthermore, due to the fact that questions covering the same set of KCs may have various difficulty levels, students perform significantly different. As shown in Figure 2, even through questions $q_2$ and $q_4$ have the same set of KCs, i.e., $c_1$ and $c_3$, students may get $q_2$ wrong but $q_4$ correct. Therefore, it is unrealistic to treat every KC in the expanded KC-response sequence identical. Inspired by the very classic and simple Rasch model in psychometrics that explicitly uses a scalar to characterize the latent factor of question difficulty, we choose to use a question-specific difficulty vector to capture the individual differences among questions on the same KC. More specifically, the $t$th representations of KC (i.e., $\mathbf{x}_t$) and interaction (i.e., $\mathbf{y}_t$) in the expanded KC sequence of concept $c_k$ are represented as follows:

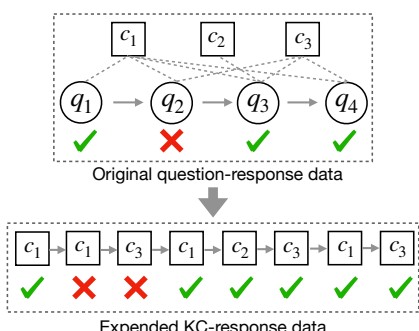

Figure 2: Graphical illustration of transforming the original question-response data into KC-response data.

$$\mathbf{x}_t = \mathbf{z}_{c_k} \oplus \mathbf{m}_{q_j} \odot \mathbf{v}_{c_k}; \quad \mathbf{y}_t = \mathbf{z}_{c_k} \oplus \mathbf{r}_{q_j}; \quad \mathbf{z}_{c_k} = \mathbf{W}_c \cdot \mathbf{e}_{c_k}; \quad \mathbf{r}_{q_j} = \mathbf{W}_q \cdot \mathbf{e}_{q_j}$$

where $\mathbf{z}_{c_k}$ denotes the latent representation of the KC $c_k$. $\mathbf{m}_{q_j}$ denotes the difficulty vector of question $q_j$ and question $q_j$ contains the KC $c_k$. $\mathbf{v}_{c_k}$ represents the question-centric variation of $q_j$ covering this KC $c_k$. $\mathbf{r}_{q_j}$ denotes the representation of student response on $q_j$. $\mathbf{e}_{c_k}$ is the $n$-dimensional one-hot vector indicating the corresponding KC and $\mathbf{e}_{q_j}$ is the 2-dimensional one-hot vector indicating whether the question is answered correctly. $\mathbf{z}_{c_k}$, $\mathbf{m}_{q_j}$, $\mathbf{v}_{c_k}$ and $\mathbf{r}_{q_j}$ are $d$-dimensional learnable real-valued vectors. $\mathbf{W}_c \in \mathbb{R}^{d \times n}$ and $\mathbf{W}_q \in \mathbb{R}^{d \times 2}$ are learnable linear transformation operations. $\odot$ and $\oplus$ are the element-wise product and addition operators. $n$ is the total number of KCs.

### 3.2.2 PREDICTION WITH ORDINARY DOT-PRODUCT ATTENTION.

Different from many existing DLKT approaches that use sophisticated neural components to model student learning and/or forgetting behaviors, we choose to use the ordinary dot-product attention function to explore and extract knowledge states from students' past learning history. Specifically, the retrieved knowledge state ($\mathbf{h}_{t+1}$) at the $(t+1)$th timestamp is computed as follows:

$$\mathbf{h}_{t+1} = \text{SelfAttention}(Q = \mathbf{x}_{t+1}, K = \{\mathbf{x}_1, \cdots, \mathbf{x}_t\}, V = \{\mathbf{y}_1, \cdots, \mathbf{y}_t\}).$$

Then we use a two-layer fully connected network to refine the knowledge state and the overall optimization function is as follows:

$$\eta_{t+1} = \mathbf{w}^\top \cdot \text{ReLU}\big(\mathbf{W}_2 \cdot \text{ReLU}(\mathbf{W}_1 \cdot [\mathbf{h}_{t+1}; \mathbf{x}_{t+1}] + \mathbf{b}_1) + \mathbf{b}_2\big) + b$$

$$\mathcal{L} = -\sum_t \big(r_t \log \sigma(\eta_t) + (1 - r_t) \log(1 - \sigma(\eta_t))\big)$$

where $\mathbf{W}_1$, $\mathbf{W}_2$, $\mathbf{w}$, $\mathbf{b}_1$, $\mathbf{b}_2$ and $b$ are trainable parameters and $\mathbf{W}_1 \in \mathbb{R}^{d \times 2d}$, $\mathbf{W}_2 \in \mathbb{R}^{d \times d}$, $\mathbf{w}, \mathbf{b}_1, \mathbf{b}_2 \in \mathbb{R}^{d \times 1}$, $b$ is scalar. $\sigma(\cdot)$ is the sigmoid function.

### 3.2.3 RELATIONSHIP TO EXISTING DLKT MODELS.

Although the proposed SIMPLEKT belongs to model category C5 discussed in Section 2, it is distinguished from attention based representative DLKT models such as AKT (Ghosh et al., 2020), SAKT

(Pandey & Karypis, 2019) and SAINT (Choi et al., 2020). The difference between SIMPLEKT and AKT are threefold: first, we omit the self-attentive question encoder and knowledge encoder in AKT and directly feed the representations of $\mathbf{x}_t$s and $\mathbf{y}_t$s into attention based knowledge state extractor; second, instead of using time decayed monotonic attention function to extract the initial knowledge state, we choose to use the ordinary dot-product function that is simple and free of hyper-parameters; third, interaction representations $\mathbf{y}_t$s are simply computed by adding representations of KCs and responses while AKT uses extra parameters to explicitly model the effects of question difficulty in interaction representations. When comparing SIMPLEKT to SAKT and SAINT, we explicitly model the latent question-centric difficulty when learning the KC representations while SAKT and SAINT ignore the question-level difference and treat all questions are identical if they contains the same set of KCs. Furthermore, SAINT adopts the encoder-decoder architecture and utilizes Transformers to model the student interaction sequence while our SIMPLEKT only uses the dot-product attention function.

## 4 EXPERIMENTS

### 4.1 DATASETS

In this paper, we experiment with 7 widely used datasets to comprehensively evaluate the performance of our models. These 7 datasets can be divided into 2 categories: (1) *D1: Datasets containing information of both questions and KCs*; and (2) *D2: Datasets containing information of either questions or KCs*. Table 1 gives real samples of question and KCs from both D1 and D2 categories. The detailed statistics of each dataset are listed in Appendix A.1.

#### 4.1.1 DATASETS CONTAINING INFORMATION OF BOTH QUESTIONS AND KCS

**ASSISTments2009 (AS2009)**[6]: This dataset is about math exercises and collected from the free online tutoring ASSISTments platform in the school year 2009-2010. It is widely used as the standard benchmark for KT methods over the last decade (Feng et al., 2009; Ghosh et al., 2020; Zhang et al., 2017). It includes 337,4115 interactions, 4,661 sequences, 17,737 questions, 123 KCs and each question has 1.1968 KCs on average.

**Algebra2005 (AL2005)**[7]: This dataset stems from KDD Cup 2010 EDM Challenge, including the detailed step-level student responses to the mathematical problems (Stamper et al., 2010). Similar to (Choffin et al., Ghosh et al., 2020; Zhang et al., 2017), a unique question is constructed by concatenating the problem name and step name. It has 884,098 interactions, 4,712 sequences, 173,113 questions, 112 KCs and the average KCs is 1.3521.

**Bridge2006 (BD2006)**[7]: This dataset is also from the KDD Cup 2010 EDM Challenge and its unique question construction is similar to the process used in Algebra2005. The dataset has 1,824,310 interactions, 9,680 sequences, 129,263 questions, 493 KCs and the average KCs is 1.0136.

**NIPS34**[8]: This dataset is provided by NeurIPS 2020 Education Challenge which contains students' answers to mathematics questions from Eedi. We use the dataset of Task 3 & Task 4 to evaluate our models (Wang et al., 2020b). There are 1,399,470 interactions, 9,401 sequences, 948 questions, 57 KCs, each question has 1.0137 KCs on average.

#### 4.1.2 DATASETS CONTAINING INFORMATION OF EITHER QUESTIONS OR KCS

**Statics2011**[9]: This dataset is collected from an engineering statics course taught at the Carnegie Mellon University during Fall 2011 (Steif & Bier, 2014). Its unique question construction is similar to the process used in Algebra2005. The dataset has 189,292 interactions, 1,034 sequences and 1,223 questions.

---

[6] https://sites.google.com/site/assistmentsdata/home/2009-2010-assistment-data/skill-builder-data-2009-2010.

[7] https://pslcdatashop.web.cmu.edu/KDDCup/.

[8] https://eedi.com/projects/neurips-education-challenge

[9] https://pslcdatashop.web.cmu.edu/DatasetInfo?datasetId=507

**ASSISTments2015 (AS2015)**[10]: Similar to ASSISTments2009, this dataset is collected from the ASSISTments platform in the year of 2015, and it has the largest number of students among the other ASSISTments datasets. It ends up with 682,789 interactions, 19,292 sequences and 100 KCs after pre-processing.

**POJ**[11]: This dataset is collected from Peking coding practice online platform and provided by Pandey & Srivastava (2020). It has 987,593 interactions, 20,114 sequences and 2,748 questions.

Following the data pre-processing steps suggested by (Liu et al., 2022), we remove student sequences shorter than 3 attempts and set the maximum length of student interaction history to 200 for a high computational efficiency.

Table 1: Examples of questions and KCs from D1 and D2.

| Category | Dataset | Question | Knowledge Components |
|---|---|---|---|
| D1 | NIPS34 | Which calculation is incorrect? A.(-7)*2=-14 B.(-7)*(-2)=-14 C.7*2=14 D. 7*(-2)=-14 | Multiplying and Dividing Negative Numbers |
| D1 | NIPS34 | Which of the following number is a factor of 60 and a multiple of 6 ... A.3 B.12 C.20 D.120 | Factors and Highest Common Factor Multiples and Lowest Common Multiple |
| D2 | POJ | Given 2 equations on the variables x and y, solve for x and y. | Not Available |
| D2 | POJ | Given a big integer number, you are required to find out whether it's a prime number. | Not Available |

### 4.2 BASELINES

To comprehensively and systematically evaluate the performance of SIMPLEKT, we compare SIMPLEKT against 12 DLKT baseline models from aforementioned 5 categories in Section 2 as follows:

- **C1: DKT** (Piech et al., 2015): directly uses RNNs to model students' learning processes.

- **C1: DKT+** (Yeung & Yeung, 2018): improves DKT by addressing the reconstruction and inconsistent issues.

- **C1: DKT-F** (Nagatani et al., 2019): improves DKT by considering students' forgetting behaviors.

- **C1: KQN** (Lee & Yeung, 2019): utilizes the dot product of the students' ability and KC representations to predict student performance.

- **C1: LPKT** (Shen et al., 2021): designs the learning cell to model students' learning processes.

- **C1: IEKT** (Long et al., 2021): estimates student knowledge state via the student cognition and knowledge acquisition estimation modules.

- **C2: DKVMN** (Zhang et al., 2017): exploits the relationships among KCs and estimate student mastery via memory networks.

- **C3: ATKT** (Guo et al., 2021): uses adversarial perturbations to enhance the generalization of the attention-LSTM based KT model.

- **C4: GKT** (Nakagawa et al., 2019): casts the knowledge structure as a graph and reformulate the KT task as a node-level classification problem.

- **C5: SAKT** (Pandey & Karypis, 2019): uses self-attention to identify the relevance between the interactions and KCs.

- **C5: SAINT** (Choi et al., 2020): a Transformer-based model for KT that encode exercise and responses in the encoder and decoder respectively.

- **C5: AKT** (Ghosh et al., 2020): models forgetting behaviors during the relevance computation between historical interactions and target questions.

### 4.3 EXPERIMENTAL SETUP

Similar to (Liu et al., 2022), we randomly withhold 20% of the students' sequences for model evaluation and we perform standard 5-fold cross validation on the rest 80% of each dataset. We select ADAM (Kingma & Ba, 2014) as the optimizer to train our model. The maximum of the training epochs is set to 200, and an early stopping strategy is used to speed up the training process. The embedding dimension, the hidden state dimension, the two dimension of the prediction layers are set to [64, 128], the learning rate and dropout rate are set to [1e-3, 1e-4, 1e-5] and [0.05, 0.1, 0.3, 0.5] respectively, the number of blocks and attention heads are set to [1, 2, 4] and [4, 8], the seed

---

[10]https://sites.google.com/site/assistmentsdata/datasets/2015-assistments-skill-builder-data

[11]https://drive.google.com/drive/folders/1LRljqWfODwTYRMPw6wEJ_mMt1KZ4xBDk

is set to [42, 3407] for reproducing the experimental results. Our model is implemented in PyTorch and trained on NVIDIA RTX 3090 GPU device. Similar to all existing DLKT research, we use the AUC as the main evaluation metric, and use accuracy as the secondary metric.

## 4.4 EXPERIMENTAL RESULTS

**Overall Performance**. Table 2 and Table 3 summarize the overall prediction performance of SIM-PLEKT and all baselines in terms of the average AUC and accuracy scores. Marker ∗, ○ and ● indicates whether SIMPLEKT is statistically superior/equal/inferior to the compared method (using paired t-test at 0.01 significance level). The last column shows the total number of win/tie/loss for SIMPLEKT against the compared method on all 7 datasets (e.g., #win is how many times SIMPLEKT significantly outperforms that method).

Table 2: Overall AUC performance of SIMPLEKT and all baselines. "-" indicates the method is inapplicable for that dataset.

| Model | D1: Datasets containing info. of both questions and KCs | | | | D2: Datasets containing info. of either questions or KCs | | | SIMPLEKT |
|---|---|---|---|---|---|---|---|---|
| | AS2009 | AL2005 | BD2006 | NIPS34 | Statics2011 | AS2015 | POJ | #win/#tie/#loss |
| DKT | 0.7541±0.0011* | 0.8149±0.0011* | 0.8015±0.0008* | 0.7689±0.0002* | 0.8222±0.0013● | 0.7271±0.0005● | 0.6089±0.0009* | 5/0/2 |
| DKT+ | 0.7547±0.0017* | 0.8156±0.0011* | 0.8020±0.0004* | 0.7696±0.0002* | 0.8279±0.0004● | 0.7285±0.0006● | 0.6173±0.0007* | 5/0/2 |
| DKT-F | - | 0.8147±0.0013* | 0.7985±0.0013* | 0.7733±0.0003* | 0.7839±0.0061* | - | 0.6030±0.0023* | 5/0/0 |
| KQN | 0.7477±0.0011* | 0.8027±0.0015* | 0.7936±0.0014* | 0.7684±0.0003* | 0.8232±0.0007● | 0.7254±0.0004● | 0.6080±0.0015* | 5/0/2 |
| LPKT | 0.7814±0.0022● | 0.8274±0.0014● | 0.8055±0.0006* | 0.8035±0.0003○ | - | - | - | 1/1/2 |
| IEKT | 0.7861±0.0027● | 0.8416±0.0014● | 0.8125±0.0009* | 0.8045±0.0002● | - | - | - | 1/0/3 |
| DKVMN | 0.7473±0.0006* | 0.8054±0.0011* | 0.7983±0.0009* | 0.7673±0.0004* | 0.8093±0.0017* | 0.7227±0.0004* | 0.6056±0.0022* | 7/0/0 |
| ATKT | 0.7470±0.0008* | 0.7995±0.0023* | 0.7889±0.0008* | 0.7665±0.0001* | 0.8055±0.0020* | 0.7245±0.0007* | 0.6075±0.0012* | 7/0/0 |
| GKT | 0.7424±0.0021* | 0.8110±0.0009* | 0.8046±0.0008* | 0.7689±0.0024* | 0.8040±0.0065○ | 0.7258±0.0012● | 0.6070±0.0036* | 5/1/1 |
| SAKT | 0.7246±0.0017* | 0.7880±0.0063* | 0.7740±0.0008* | 0.7517±0.0005* | 0.7965±0.0014* | 0.7114±0.0003* | 0.6095±0.0013* | 7/0/0 |
| SAINT | 0.6958±0.0023○ | 0.7775±0.0017* | 0.7781±0.0013* | 0.7873±0.0007* | 0.7599±0.0139* | 0.7026±0.0011* | 0.5563±0.0012* | 6/1/0 |
| AKT | 0.7853±0.0017● | 0.8306±0.0019● | 0.8208±0.0007● | 0.8033±0.0003* | 0.8309±0.0009● | 0.7281±0.0004● | 0.6281±0.0013● | 1/0/6 |
| simpleKT | 0.7744±0.0018 | 0.8254±0.0003 | 0.8160±0.0006 | 0.8035±0.0000 | 0.8199±0.0011 | 0.7248±0.0005 | 0.6252±0.0005 | - |

Table 3: Overall Accuracy performance of SIMPLEKT and all baselines. "-" indicates the method is inapplicable for that dataset.

| Model | D1: Datasets containing info. of both questions and KCs | | | | D2: Datasets containing info. of either questions or KCs | | | SIMPLEKT |
|---|---|---|---|---|---|---|---|---|
| | AS2009 | AL2005 | BD2006 | NIPS34 | Statics2011 | AS2015 | POJ | #win/#tie/#loss |
| DKT | 0.7244±0.0014* | 0.8097±0.0005* | 0.8553±0.0002* | 0.7032±0.0004* | 0.7969±0.0006● | 0.7503±0.0003* | 0.6328±0.0020* | 5/0/2 |
| DKT+ | 0.7248±0.0009* | 0.8097±0.0007● | 0.8553±0.0003* | 0.7039±0.0004* | 0.7977±0.0006● | 0.7510±0.0004● | 0.6482±0.0021* | 4/0/3 |
| DKT-F | - | 0.8090±0.0005● | 0.8536±0.0004* | 0.7076±0.0002* | 0.7872±0.0011* | - | 0.6371±0.0030* | 4/0/1 |
| KQN | 0.7228±0.0009* | 0.8025±0.0006* | 0.8532±0.0006* | 0.7028±0.0001* | 0.7978±0.0007● | 0.7500±0.0003* | 0.6435±0.0017* | 6/0/1 |
| LPKT | 0.7355±0.0015● | 0.8145±0.0007● | 0.8544±0.0008* | 0.7341±0.0003● | - | - | - | 1/0/3 |
| IEKT | 0.7375±0.0042● | 0.8236±0.0010● | 0.8553±0.0023* | 0.7330±0.0002● | - | - | - | 1/0/3 |
| DKVMN | 0.7199±0.0010* | 0.8027±0.0007* | 0.8545±0.0002* | 0.7016±0.0005* | 0.7929±0.0006* | 0.7508±0.0006○ | 0.6393±0.0015* | 6/1/0 |
| ATKT | 0.7208±0.0009* | 0.7998±0.0019* | 0.8511±0.0004* | 0.7013±0.0002* | 0.7904±0.0011* | 0.7494±0.0002* | 0.6332±0.0023* | 7/0/0 |
| GKT | 0.7153±0.0032* | 0.8088±0.0008● | 0.8555±0.0002* | 0.7014±0.0028* | 0.7902±0.0021○ | 0.7504±0.0010* | 0.6117±0.0147* | 5/1/1 |
| SAKT | 0.7063±0.0018* | 0.7954±0.0020* | 0.8461±0.0005* | 0.6879±0.0004* | 0.7879±0.0015* | 0.7474±0.0002* | 0.6407±0.0035* | 7/0/0 |
| SAINT | 0.6936±0.0034○ | 0.7791±0.0016* | 0.8411±0.0065* | 0.7180±0.0006* | 0.7682±0.0056* | 0.7438±0.0010* | 0.6476±0.0003* | 6/1/0 |
| AKT | 0.7392±0.0021● | 0.8124±0.0011● | 0.8587±0.0005● | 0.7323±0.0005* | 0.8021±0.0011● | 0.7521±0.0005● | 0.6492±0.0010* | 2/0/5 |
| simpleKT | 0.7320±0.0012 | 0.8083±0.0005 | 0.8579±0.0003 | 0.7328±0.0001 | 0.7957±0.0020 | 0.7508±0.0004 | 0.6522±0.0008 | - |

From Table 2 and Table 3, we find the following results: (1) compared to other baseline methods, SIMPLEKT almost always ranks top 3 in terms of AUC scores and achieves 55 wins, 3 ties and 18 loss in total against 12 baselines on 7 public datasets of different domains. This indicates the strength of SIMPLEKT as a baseline of KT; (2) in general, the SIMPLEKT approach performs better on D1 datasets that have both question and KC information available. When training SIMPLEKT on D2 datasets, due to the lack of distinguished information about questions and KCs, the explicit question-centric difficulty modeling degrades and the KC representations become the question agnostic; (3) when comparing SIMPLEKT to other attentive models, i.e., SAKT, SAINT and AKT, in C5 category, our SIMPLEKT beats SAKT and SAINT on all the datasets, which indicates the effectiveness of explicit question-centric difficulty modeling. Although our SIMPLEKT approach is significantly worse than the AKT approach on 6 datasets, the performance gaps are quite minimal that are mostly within a 0.5% range. On the other hand, SIMPLEKT is much more concise compared to the two-layer attentive architecture in the AKT approach; (4) the SIMPLEKT outperforms many deep sequential models in category C1, including DKT, DKT+, DKT-F, KQN on AS2009, AL2005, BD2006, NIPS34 and POJ. We believe this is because the above 4 sequential models use KCs to index questions cannot capture the individual differences among questions with the same KCs which is crucial to predict student future performance; (5) comparing SIMPLEKT and IEKT, we can see, IEKT has better prediction performance on AS2009, AL2005, and NIPS34. This is because IEKT captures both question-level and KC-level variations in its representations and at the same time, it designs two specific neural modules to estimate individual cognition and acquisition

abilities; (6) compare to other different types of models in C2, C3, and C4, such as DKVMN, ATKT and GKT, our SIMPLEKT achieves better performance by using an ordinary dot-product attention function without any memory mechanism, adversarial learning or graph constructions, which encourages the further educational researchers and practitioners to develop effective models with a design of simplicity.

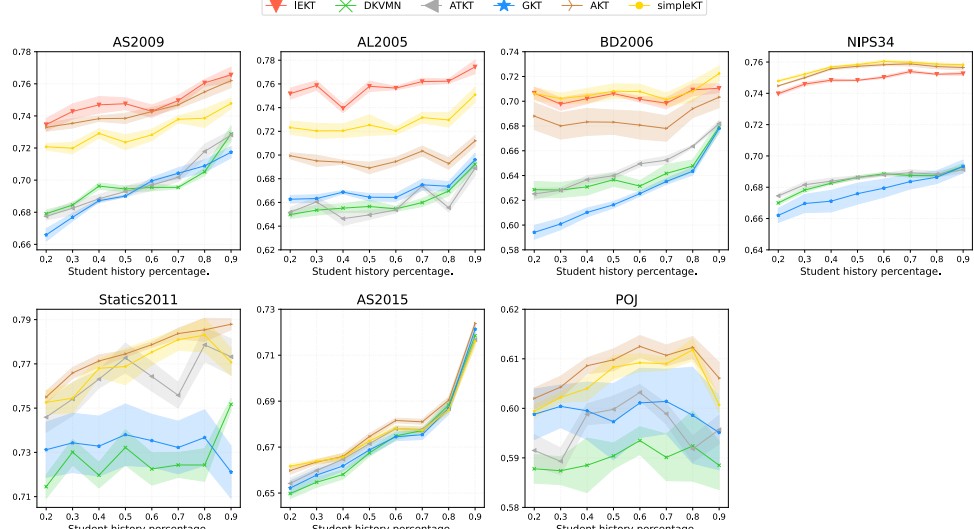

Figure 3: Non-accumulative predictions in the multi-step ahead scenario in terms of AUC.

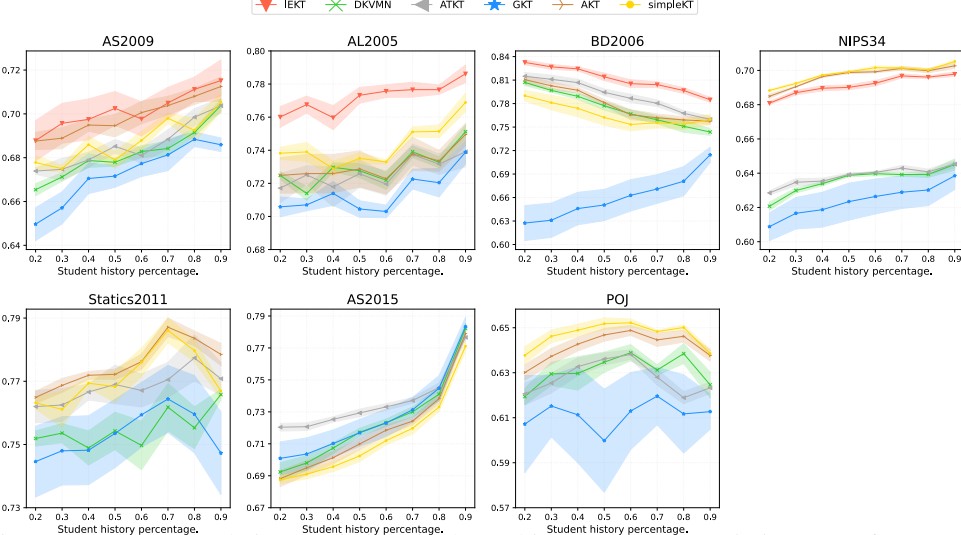

Figure 4: Non-accumulative predictions in the multi-step ahead scenario in terms of Accuracy.

**Multi-step KT Prediction Performance**. In order to make the prediction close to real application scenarios, we also predict our model in multi-step prediction which predicts a span of student's responses given the student's historical interaction sequence. Practically, accurate multi-step KT prediction will provide constructive feedback to learning path selection and construction and help teachers adaptively adjust future teaching materials. We conduct the prediction in a non-accumulative setting that predicts all future values all at once to avoid accumulative prediction errors. To have a fine-grained analysis in the multi-step ahead prediction scenario, we further experiment with DLKT models on different portions of observed student interactions. Specifically, we vary the observed percentages of student interaction length from 20% to 90% with step size of 10%. Due to the space limit, we select the best baseline in each category, i.e., IEKT, DKVMN, ATKT, GKT, AKT as the representative approaches and the results in terms of AUC and accuracy are shown in Figure 3 and Figure 4. We make the following observations: (1) with the increasing historical information, the student AUC performance prediction become more accurate in most cases, which is in line with the

real-world educational scenario; (2) the attention based model almost outperforms other KT models in Statics2011, AS2005 and POJ according to AUC scores, this is because the attention mechanism is expert in capturing the long-term dependencies; and (3) our SIMPLEKT achieves the best prediction performance in BD2006, NIPS34 in terms of AUC, which indicates the proposed method is simple yet powerful.

**Qualitative Visual Analysis**. In this section, we qualitatively show the visualization of the prediction results made by SIMPLEKT in Figure 5. To better understand the model predictive behavior, we compute the historical error rate (HER) per question from the data and use the HERs as surrogates of question difficulties. Due to the space limit, more illustrative and fine-grained results are provided in Appendix A.2. As we can see from Figure 5, when a student meets a certain KC for the first time, the higher HER of the question, the lower the probability that student will get it correct. For example, the HERs for questions $q_{527}$, $q_{509}$, $q_{512}$, $q_{518}$ and $q_{219}$ are 0.35, 0.41, 0.20, 0.51, and 0.48 and the corresponding prediction probabilities of SIMPLEKT are 0.74, 0.52, 0.84, 0.45, and 0.51 respectively. Furthermore, for those questions that cover the same set of KCs, such as questions $q_{526}$ and $q_{529}$, the model prediction probability decreases when the corresponding HER increases.

Figure 5: Visualization of a student's prediction results on SIMPLEKT.

**Ablation Study**. We systematically examine the effect of the key component of question difficulty modeling by constructing two model variants: (1) SIMPLEKT-ScalarDiff that changes the question-centric difficulty vector $\mathbf{m}_{q_t}$ to scalar; and (2) SIMPLEKT-NoDiff that completely ignores the question difficulty modeling and simply set $\mathbf{x}_t$ to $\mathbf{z}_{c_t}$. The prediction performance on all datasets that belong to D1 are reported in Table 4. Please note that since datasets in D2 only have either question information or KC information, SIMPLEKT, SIMPLEKT-ScalarDiff, and SIMPLEKT-NoDiff essentially become mathematically unidentifiable. From Table 4, we can easily observe that (1) the SIMPLEKT method outperforms the two model variants and especially when removing the question difficulty modeling component, the prediction performance decreases more than 2% on all D1 datasets. This empirically verifies the importance of question-centric difficulty modeling when making student performance prediction in KT scenarios; and (2) comparing SIMPLEKT and SIMPLEKT-ScalarDiff, the performance is very minimal. We believe this is because the KC representation $\mathbf{x}_t$ is based on the simple additive assumption and a scalar is expressive enough to represent question-level difficulty under this assumption.

Table 4: The performance of different variants in SIMPLEKT.

|  | AS2009 | AL2005 | BD2006 | NIPS34 |
|---|---|---|---|---|
| SIMPLEKT | 0.7744±0.0018 | 0.8254±0.0003 | 0.8160±0.0006 | 0.8035±0.0000 |
| SIMPLEKT-ScalarDiff | 0.7740±0.0021 | 0.8250±0.0013 | 0.8159±0.0011 | 0.8008±0.0012 |
| SIMPLEKT-NoDiff | 0.7411±0.0016 | 0.8048±0.0018 | 0.7922±0.0011 | 0.7646±0.0005 |

## 5 CONCLUSION

In this work, we propose SIMPLEKT, a simple but tough-to-beat approach to solve KT task effectively. Motivated by the Rasch model in psychometrics, the SIMPLEKT approach is designed to capture individual differences among questions with the same KCs. Furthermore, the proposed SIMPLEKT approach simplifies the sophisticated student knowledge state estimation component with the ordinary dot-product attention function. Comprehensive experimental results demonstrate that SIMPLEKT is able to beat a wide range of recently proposed DLKT models on various datasets from different domains. We believe this work serves as a strong baseline for future KT research.

## REPRODUCIBILITY STATEMENT

The code of SIMPLEKT and its variants, i.e., SIMPLEKT-ScalarDiff and SIMPLEKT-NoDiff, to reproduce the experimental results can be found at https://github.com/pykt-team/pykt-toolkit. We give the details of data-preprocessing and the training hyper-parameters of SIMPLEKT in Section 4.1 and Section 4.3. The code of the 12 comparison models is accessible from an open-sourced PYKT python library at https://pykt.org/. We choose to use the same data partitions of train, validation, test sets as PYKT and hence all the results can be easily reproducible. All the model training details of all baselines can be found at https://pykt-toolkit.readthedocs.io/en/latest/pykt.models.html.

## ACKNOWLEDGMENTS

This work was supported in part by National Key R&D Program of China, under Grant No. 2020AAA0104500; in part by Beijing Nova Program (Z201100006820068) from Beijing Municipal Science & Technology Commission; in part by NFSC under Grant No. 61877029 and in part by Key Laboratory of Smart Education of Guangdong Higher Education Institutes, Jinan University (2022LSYS003).

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
