# OpenReview forum: "simpleKT: A Simple But Tough-to-Beat Baseline for Knowledge Tracing"
_ICLR.cc/2023/Conference — ICLR 2023 poster_

### Official Review · Reviewer_vuA2 · 2022-10-23

**Confidence:** 4
**Correctness:** 3
**Technical Novelty And Significance:** 2
**Empirical Novelty And Significance:** 2
**Recommendation:** 3

**Clarity, Quality, Novelty And Reproducibility:**

Clarity: the motivations and ideas are clear in this work.

Quality: The techniques of this work is limited, directly using Transformer to solve their tasks.

Novelty: The novelty of this work is not high,

Reproducibility: The reproducibility of this work is good, since it is simple and source code is available at this time.

**Strength And Weaknesses:**

Strength

1. The paper is well written and well presented.

2. The source code is publicly avaiblale.

3. The experimental results are good, comparing to existing baselines.


Weakness
1. The technical contributions of this work are very limited, simply adopting the tranformer model for their tasks. Overall contributions are below the bar of ICLR in my mind.

**Summary Of The Paper:**

This paper proposes SIMPLEKT, a simple but tough-to-beat KT baseline that is simple to implement, computationally friendly and robust to a wide range of KT datasets across different domains.

**Summary Of The Review:**

In summary, the topic in this work is interesting, however the overall contributions of this work are not very enough for ICLR.

---

### Official Review · Reviewer_nkiF · 2022-10-25

**Confidence:** 3
**Correctness:** 3
**Technical Novelty And Significance:** 2
**Empirical Novelty And Significance:** 3
**Recommendation:** 8

**Clarity, Quality, Novelty And Reproducibility:**

Clarity: The clarity for this work is fair. It could improve with some rewrites with an eye towards the reader's experience:
    - Define "KC" in the abstract before referencing it. It is currently not defined until the Introduction.
    - The first paragraph of page 2 has a heavy mix of citations and prose. Whereas there is not much prose in the citation-heavy of paragraph of page 1 (hence being able to get away with it the first time), it is much harder to read the page 2 paragraph. It could be helpful to restate/recap the important claims to ensure they do not get get missed by the reader.
    - Although Figure 1 is very helpful for generally understanding the KT task, it is hard to follow the distinction between D1 (datasets containing info about both q's and KCs) and D2 (datasets containing info of *either* q's or KCs). One example of an instance from each type would be helpful to see.

Quality: This work is high-quality. By comprehensively surveying the literature, the authors identify a gap in approaches, which they fill using a simple model. This simple model employs tactics which are well-suited for the task (eg attention, question-difficulty info, etc) to demonstrate that many previous approaches were not as effective as once understood, despite their bells and whistles. This is a very useful finding, and it is done by measuring across 7 datasets.

Novelty: The novelty of this approach is, on the one hand, low, because it uses a simple approach. However, it demonstrates why a "simple" model is nonetheless able to perform better than nearly a dozen "more novel" models.

Reproducibility: This work is highly-reproducible. It provides a link to an anonymized repo with code and one of the datasets.

**Strength And Weaknesses:**

STRENGTHS
- This work gives a very comprehensive review of dozens of work in the KT space and how they work.
- By comparing against 12 previous models, this weighs in on disagreements in the literature about model comparisons.
- Demonstrating that such a simple model can outperform many deep learning-based approaches provides meaningful insight into the nature of the task, which can inspire future work and model designs.

WEAKNESSES
- I could not tell whether the 12 comparison models had publicly available code or were reimplemented by the authors (or something else). Clarification would be helpful.


**Summary Of The Paper:**

This work aims to address inconclusive findings in the literature about which models perform best for Knowledge Tracing (KT). In this paper, the authors present a baseline which convincingly outperforms most published work despite being simpler in nature.

**Summary Of The Review:**

In this paper, the authors present a baseline which convincingly outperforms most published work despite being simpler in nature. The baseline compares against 12 other models -- such a comprehensive comparison is itself valuable to the field. Additionally, the works conducts an ablation study, which provides important insight into understanding what makes the simple model nonetheless effective (e.g. how much is from the question-difficulty modeling?).

---

### Official Review · Reviewer_EgtS · 2022-10-25

**Confidence:** 2
**Correctness:** 4
**Technical Novelty And Significance:** 3
**Empirical Novelty And Significance:** 3
**Recommendation:** 6

**Clarity, Quality, Novelty And Reproducibility:**

This paper is easy to understand even for a reader without much background on the knowledge tracing task.


**Strength And Weaknesses:**

Strengths:
A simple-to-implement and strong baseline is constructed for the knowledge tracing (TK) task, which can facilitate future KT research.

Weaknesses:
I am not familiar with the knowledge tracing task. Thus, I cannot provide useful comments and suggestions to improve this paper.


**Summary Of The Paper:**

This paper builds a simple yet strong baseline, named SimpleKT, for knowledge tracing (KT). The newly introduced baseline is motivated by the Rasch model in psychometrics to capture the individual differences in questions of the same KC. It only contains a few simple operations, i.e., element-wise product, addition, and the ordinary dot-product attention, which is simple to implement. Surprisingly, the proposed simple method can beat most of the 12 representative baselines on 7 public datasets.

**Summary Of The Review:**

A simple-to-implement and well-motivated baseline model is presented for the Knowledge tracing task, which wins most of 12 existing methods on 7 public datasets. The newly constructed baseline can facilitate future research on KT.

---

### Decision · Program_Chairs · 2023-01-20

**Decision:**

Accept: poster

**Justification For Why Not Higher Score:**

This work would benefit a lot from poster discussion.

**Justification For Why Not Lower Score:**

N/A

**Metareview: Summary, Strengths And Weaknesses:**

This work introduces a simple and strong knowledge tracking baseline. As reviewers pointed out, this approach is simple, computationally friendly, and performs better than many existing approaches on a wide range of tasks across different domains.  The paper itself is well-written. The concern was around whether the technical contribution is novel enough. After the discussion among reviewers and reviewing authors’ responses, I’m leaning towards acceptance as the simplicity of a method (e.g., utilizing transformers for this task) is not a weakness and strong baselines help us understand the task itself in many ways.



**Note From Pc:**

if the above contains the word "oral" or "spotlight" please see: "oral" presentation means -> notable-top-5% and "spotlight" means -> notable-top-25%. As stated in our emails, we are disassociating presentation type from AC recommendations

**Summary Of Ac-Reviewer Meeting:**

N/A